# Efficiency of *Lactiplantibacillus plantarum* JT-PN39 and *Paenibacillus motobuensis* JT-A29 for Fermented Coffee Applications and Fermented Coffee Characteristics

**DOI:** 10.3390/foods12152894

**Published:** 2023-07-29

**Authors:** Teerawat Ngamnok, Wutigri Nimlamool, Daniel Amador-Noguez, Tanapat Palaga, Jomkhwan Meerak

**Affiliations:** 1Master’s Degree Program in Applied Microbiology, Department of Biology, Faculty of Science, Chiang Mai University, Chiang Mai 50200, Thailand; teerawat_ng@cmu.ac.th; 2Department of Biology, Faculty of Science, Chiang Mai University, Chiang Mai 50200, Thailand; 3The Graduate School, Chiang Mai University, Chiang Mai 50200, Thailand; 4Department of Pharmacology, Faculty of Medicine, Chiang Mai University, Chiang Mai 50200, Thailand; wutigri.nimlamool@cmu.ac.th; 5Department of Bacteriology, University of Wisconsin—Madison, Madison, WI 53706, USA; amadornoguez@wisc.edu; 6Department of Microbiology, Faculty of Science, Chulalongkorn University, Bangkok 10330, Thailand; tanapat.p@chula.ac.th; 7Center of Excellence in Materials Science and Technology, Chiang Mai University, Chiang Mai 50200, Thailand

**Keywords:** fermented coffee, *Lactiplantibacillus plantarum*, *Paenibacillus motobuensis*, enzyme

## Abstract

To develop a process for low-cost and ecologically friendly coffee fermentation, civet gut bacteria were isolated and screened to be used for fermentation. Among 223 isolates from civet feces, two bacteria exhibited strong protease, amylase, lipase, pectinase, and cellulase activities. By analyzing 16S rDNA phylogeny, those bacteria were identified to be *Lactiplantibacillus plantarum* JT-PN39 (LP) and *Paenibacillus motobuensis* JT-A29 (PM), where their potency (pure or mixed bacterial culture) for fermenting 5 L of arabica parchment coffee in 48–72 h was further determined. To characterize the role of bacteria in coffee fermentation, growth and pH were also determined. For mixed starter culture conditions, the growth of PM was not detected after 36 h of fermentation due to the low acid conditions generated by LP. Coffee quality was evaluated using a cupping test, and LP-fermented coffee expressed a higher cupping score, with a main fruity and sour flavor, and a dominant caramel-honey-like aroma. Antioxidant and anti-foodborne pathogenic bacteria activity, including total phenolic compounds of PM and LP fermented coffee extracts, was significantly higher than those of ordinary coffee. In addition, LP-fermented coffee expressed the highest antibacterial and antioxidant activities among the fermented coffee. The toxicity test was examined in the murine macrophage RAW 264.7 cell, and all fermented coffee revealed 80–90% cell variability, which means that the fermentation process does not generate any toxicity. In addition, qualifications of non-volatile and volatile compounds in fermented coffee were examined by LC-MS and GC-MS to discriminate the bacterial role during the process by PCA plot. The flavors of fermented coffee, including volatile and non-volatile compounds, were totally different between the non-fermented and fermented conditions. Moreover, the PCA plot showed slightly different flavors among fermentations with different starter cultures. For both the cupping test and biological activities, this study suggests that LP has potential for health benefits in coffee fermentation.

## 1. Introduction

Coffee is one of the most consumed beverages in the world, and the need to develop new varieties, unique characteristics, and increase the commodity has been considered. The difference in coffee processing, from harvesting to cupping to the cultivation regions, reflects the variety of its quality and popularity [1]. In addition, coffee processing has also brought undesired side effects since coffee production and consumption have generated impressive quantities of related wastes and environmental pollution. To increase the product’s economic value, specialty coffee production by animal feeding and gut fermentation are globally known to create unique aromas and flavors that lead to a higher product price. However, due to issues related to animal cruelty in laboratories, many reports have focused on replacing animal utilization with microbial fermentation [2,3]. The mixed culture of bacteria or bacteria and yeast were described in many previous studies for their ability to generate animal fermentation-like coffee or increase coffee market profitability. Regarding the preliminary examination of fermented coffee’s chemical properties, the fermented prototypes are commonly created at a laboratory scale in amounts of 500 milliliters to 2 L within a period of 7–30 days. Vaast et al. [4] reported that the use of bacterial inoculum resulted in physiological changes during fermentation. Those included a decrease in the level of general sugars and fatty acids that are responsive to the generation of aroma and flavors. Most processes mentioned previously were regarded as coffee mucilage digestion and removal. During long fermentation, microbial growth and their enzyme activities in digesting coffee, or regarding certain synthetic pathways in coffee beans, are less controlled, making it difficult to maintain the repeatable process of quality control. Furthermore, long fermentation periods can generate greater waste and undesired bacterial products, with an ineffective cost for further industrial scaling. In addition to the potency of selected inoculum, the normal microbial flora of coffee fruit is dependent upon the weather and cultivation area, which contributes to the unmanageability of popular characteristics. Currently, in Columbian fermented coffee processing, next-generation sequencing is used to investigate bacterial and fungal communities, and 160 genera of bacterial diversity associated with spontaneous coffee-bean fermentation have been found. Among those, lactic acid bacteria, with the two dominant *Leuconostoc* and *Lactiplantibacillus*, were identified to be the main population. In contrast, fungal groups were presented as a homogeneous dominant in the genus *Pichia* [5].

Current research indicates that the coffee fermentation process is becoming more popular for producing specialty coffee based on sensorial profiles, but it is necessary to establish a standard process that guarantees the certainty of an industrial-scale application [6]. For instance, one aspect of increasing the final value of the product is its beneficial impact on the consumer’s health. Endogenous enzymes in green beans and microorganisms can digest macromolecules to regenerate some active compounds that stimulate the immune function and prevent some diseases. Its consumption has been historically related to preventing and lowering the risk of type 2 diabetes mellitus, obesity, cardiovascular disease, and some types of cancer [7]. Thus, the selection of microorganisms for the fermentation step is a crucial process.

Moreover, non-dairy health benefit products utilize probiotics as the starter for the fermentation process and have been illustrated by many studies for future promotion as functional foods or beverages [8,9]. Küçükgöz et al. [10] described the uses of *L. acidophilus*, *L. casei*, *L. delbrueckii*, and *L. plantarum* to ferment tomato juice to improve its quality and increase its biological properties as a functional juice. It was suggested that plant-based probiotic products were considered the most suitable and acceptable plant food medium for probiotic bacteria. However, only a few of the probiotics were supported for use in coffee fermentation.

In this study, a single bacterium inoculum and a mixed strain were evaluated for their ability to establish ecologically friendly and low-cost industrially fermented Arabica coffee production in Northern Thailand. The inoculum was recruited from civet’s stool and investigated for their target enzymes for the production of volatile compounds giving rise to the unique popular aroma, physical and chemical changes, and the lack of production of undesirable or toxic metabolites. The starter cultures are useful for assuring better fermentation control and predictability of the final product [11]. In this regard, fermented coffee also demonstrated the benefit of the new beverage processing to human health. In addition, this research aimed to offer an alternative method to avoid utilizing civets and support the animal ethical trading barrier in several regions.

## 2. Materials and Methods

### 2.1. Bacterial Isolation and Screening for Enzyme Production

Bacteria were isolated from various sources, such as civet stool, cow dung and mango peel. Based on the macromolecule compositions and pectin-rich silver skin of green coffee beans capable of being digested and regenerating chemical changes, cellulolytic enzymes, pectinase, amylase, protease, and lipase were the main target enzymes for isolation and screening of bacteria. Fresh animal feces were received from local coffee farmers in Chiang Rai and Lampang province, while ripened mangoes were purchased from the markets in Chiang Mai during 2018–2019. The enrichment was carried out with 1 g of sample dissolved in each enzyme substrate (1% *w*/*v* of carboxymethyl cellulose (CMC), pectin, soluble starch, skim milk, and tween 80, respectively) and then 10 mL of minimal media (M9) was added, and the mixture was left for 1 week at 37 °C. The positive samples were diluted and plated on the same substrate (1% *w*/*v*), and then DeMan, Rogosa, and Sharpe (MRS) agar or M9 agar was added [12]. The ability for hydrolysis was investigated by Congo red dye for cellulolytic, 50 mM potassium iodide solution flooding for pectinase, iodine solution for amylase, and the clear zone surrounded coolly for protease and lipase, respectively. The positive colonies of each enzyme that appeared between 24 and 72 h after incubation at 37 °C were collected for further analysis.

In addition, colonies that showed a greater enzyme activity were chosen to test the ability of the remaining enzyme production to underly the capability of multi-enzymes producing strains for coffee fermentation. Selected bacteria were repeatedly determined for the hydrolysis activity in 1% (*w*/*v*) substrate added to 100 mL culture broth at 37 °C for 24 h. Determination of cellulase, pectinase, and amylase activity was performed by colorimetric measurements of sugar reduction following the DNS method as described by Jain et al. [13]. The non-specific protease activity was determined using casein as a substrate based on a standardized procedure mentioned by Kumari et al. [14], and the proteolytic amino acids were determined using Lowry method. Lipase activity was quantified using ρ-nitrophenyl palmitate (ρ-NPP) as a substrate based on the method by Gupta et al. [15]. The measurement of 1 unit/mL of non-specific enzyme activity was calculated for a description of the number of enzymes that catalyze the reaction of 1 nmol of the substrate per minute per milliliter. The strains that showed the highest enzyme activity were selected for the next examination for their ability to digest parchment coffee. The bacteria were maintained in MRS or tryptic soy broth (TSB) plus 40% glycerol as a stock culture at −20 °C.

### 2.2. Bacterial Identification

#### 2.2.1. Genomics DNA Preparation and PCR Amplification

The first 10 isolates that expressed the highest positive test for multi-enzyme production were selected for further identification based on 16S rDNA sequences. Bacterium was cultured in 5 mL of tryptic soy broth (TSB) for 24 h at 37 °C, with or without shaking, depending on the optimal growth. Cells were harvested by centrifugation at 10,000 rpm, 4 °C for 2 min, and genomic DNA was extracted with Fungal/Bact DNA extraction kit (Zymo Research, Irvine, CA, USA) following the manufacturer’s protocol. The quantification of gDNA was measured by Nanodrop (Thermo Scientific, Waltham, MA, USA) and 1% (*w*/*v*) agarose gel electrophoresis. PCR amplification of 16S rDNA sequences was carried out using universal primers: 27F 5′-AGAGTTTGATCCTGGCTCAG-3′) and 1492R 5′-TACGGTTACCTTGTTACGACTT-3′ [16] The PCR mixture was carried out in 50 μL reaction that contained 100 ng of template DNA, Emerald Taq DNA polymerase, and 0.25 μM of each primer. The nuclease-free water was added to adjust the total volume of 50 μL. The mixture was heated to denature genomic DNA at 96 °C for 5 min and subjected to 35 cycles of 96 °C for 1 min, annealing at 58 °C for 30 s, and extension step at 72 °C for 1.5 min. The final step of DNA extension was performed at 72 °C for 10 min [16]. The amplicons were run on 1% agarose gel, and an approximation of a single 1500 bp DNA band was excised and purified using a Gel Extraction kit (Qiagen, Valencia, CA, USA) for DNA sequencing.

#### 2.2.2. DNA Sequencing and Phylogenetic Analysis

DNA sequencing was carried out using an automatic DNA sequencer (ABI PRISM 3130xl Genetic Analyzer) with an Applied Biosystems BigDye (ver. 3.1) kit. The sequences were preliminary qualified by BioEdit 7 [17] and blasted with references for the 16S rDNA database in NCBI (http://www.ncbi.nlm.nih.gov, accessed on 10 May 2022) with highly similar sequences. The 16S rDNA sequences of the type strains of the target bacteria were selected and retrieved from EzTaxon server (http://www.eztaxon.org*/*, accessed on 11 May 2022) [18]. A multiple sequence alignment was performed using ClustalX 1.83 [19,20] with the optimized alignment parameters. Then, MEGA 12 was used for phylogenic tree construction of neighbor-joining (NJ) with unambiguously aligned nucleotides. *Bacillus subtilis* DMS10 was used as the outgroup for lactic acid bacteria, and *Lactiplantibacillus plantarum* CIP03151 was used as the outgroup for other bacterial groups. NJ analysis was performed using the maximum composite likelihood method and 1000 bootstrap replications [21].

### 2.3. Examination of Coffee Digestion

The non-pathogenic isolates, capable of multi-enzyme production raking at top 5, were selected for an enzyme activity test in minimal media supplemented with 1% (*w*/*v*) parchment coffee in 100 mL M9 medium as a secondary screening. A single bacterial colony was inoculated into 10 mL MRS or TSB and incubated at 37 °C for 24 h for activation. Bacterial cells were harvested from 1 to 3 mL MRS broth or TSB via centrifugation, and the OD_600_ was adjusted to 0.1 to adjust the equal starter culture for hydrolysis of coffee beans. Then, cell suspension was transferred to 100 mL minimal medium containing 1 g of parchment coffee and cultivated at 37 °C. The total activity of enzymes was detected in culture supernatant collected at 0, 48, 96, 144, and 192 h at 37 °C, using the method described previously. Three replicates of each experiment were performed in separated culture flasks, and uninoculated was set as a control.

### 2.4. Mini-Scale Coffee Fermentation

Inoculum and process for fermented coffee production in a small 5L scale was designed based on the production of bacterial enzymes and consumer safety in accordance with US-FDA. A selection of mixed bacteria cultures or individual strains were activated in 5 mL bacterial enrichment medium mentioned above at 37 °C for 18–24 h. Then, 5% seed culture (*v*/*v*) was transferred to 100 mL of the same medium and culture condition. Total cell suspension was transferred to the closed 5 L glass reactor containing parchment coffee in a sterile M9 medium at approximately 50% for fermentation. The reactor was stored at RT for 2 to 3 days, depending on the single strain or mixed culture fermentation process that was applied. Fermented coffee beans were harvested by filtration using a filter cloth and washed with sterile distilled water once before air drying until the moisture reached 10–12% when measured by a coffee kett moisture meter (KETT PM-450, Kett Electric Laboratory, Tokyo, Japan). The green bean coffee was prepared freshly and kept at RT until the roasting process. The uninoculated conditions of each fermented formula with similar processes were set as the control, and parchment coffee used in all experiments was selected from the same cultivation area.

### 2.5. Sensory Analysis Based on SCAA Cupping Test

Fermented coffee sensory analysis was examined by one certified specialty coffee tester and 5 Q-graders as the standard protocol based on the Specialty Coffee Association of America Cupping Protocols: SCAA (SCAA, 2015) [22] by HillKoff Co., Ltd. in Chiang Mai, Thailand, the certified coffee manufacturer in Chiang Mai province, Thailand. Five hundred grams of bacteria-fermented, uninoculated, and ordinary coffee were roasted with the industrial scale machine, Silon ZR-7 (Coffee-tech engineering, Matsli’ah, Israel) at full city level (200–220 °C) for approximately 11–12 min and allowed to degas for 72 h before further analysis. Each coffee sample was evaluated using 10 g of ground coffee per 200 mL of hot water/cup. At least 5 cups of each sample were examined for fragrance/aroma, flavor, acidity, body, balance, aftertaste, uniformity, sweetness, clean cup, and overall impression. The evaluated sensory attributes were grouped into “subjective” and “objective” categories in which the “subjective” attributes were a synergistic combination of flavor, aftertaste, acidity, body, aftertaste, and overall impression, and they were quality scored according to a scale of 6–10 points with 0.25 point increments. The “objective” category included uniformity, sweetness, and clean cup, and the score was considered on a scale from 0 to 10 points, with 2 points awarded. Furthermore, the characteristic of flavors was described for each coffee [23]. In addition, an average score above 7.5 was considered to be the coffee specialty [24].

### 2.6. Coffee Extract Preparation and Biological Property Examinations

One kilogram of green coffee beans was roasted at 200–220 °C at different times to generate a light roast, medium roast, and dark roast level with a medium drum roaster and degassed, as mentioned previously. Then, coffee was finely ground using an electronic grinder. Hot brew extraction was made using the same protocol as usual daily cupping but with a 4 times higher concentration by using 10 g ground coffee in 50 mL distilled water at 90 °C for 20 min. Then, the supernatant was collected by centrifugation at 8000 rpm at 4 °C for 5 min, and the supernatant was lyophilized. The extracts were solubilized in sterile distilled water and filtrated through 0.2 µm nitrocellulose membrane for further biological property tests, and each experiment was carried out in three replications.

#### 2.6.1. Total Phenolic Compounds and Antioxidant Activity

The total phenol content (TPC) of each coffee extract with a concertation of 500 mg/mL was measured using the modification method by Andreou et al. [25]. Briefly, 5 μL of crude extract was mixed with 100 μL of 10% Folin–Ciocalteu reagent and incubated at RT for 10 min. Then, 80 μL 7.5% sodium carbonate (Na_2_CO_3_) was added to the mixture. The mixture was incubated at room temperature for 30 min, and the reaction absorbance was measured at 765 nm using a spectrophotometer (U-2900, Hitachi High-Tech Corporation, Tokyo, Japan). Gallic acid solutions at a concentration of 0 to 10 mg/mL were used to generate a standard curve. TPC results were expressed as mg gallic acid equivalent/g (mg GAE/g) of coffee extract. For the antioxidant activity, each coffee was prepared at various concentrations using ABTS radical scavenging assay. In this study, the ABTS assay has a bigger advantage than the DPPH assay in the examination of the fermented coffee extracts as it can be used at different pH values [26] and is applicable with compounds that have a lipophilic or hydrophilic character of different coffee varieties. The assay principle is based on the utilized free radical, mono-cation of 2,2′-azino-bis 3-ethylbenzothiazoline-6-sulphonic acid (ABTS), which is generated when ABTS substrate is oxidized with potassium persulfate, and the color of ABTS was measured by absorption spectra at OD 734 nm. ABTS^•+^ was prepared by reacting a 7 mM ABTS solution with a 2.45 mM potassium persulfate; then, the solution was stored in the dark at RT for 16 h. Prior to the assay, the solution was diluted with phosphate-buffered saline (PBS, pH 7.4) to obtain an absorbance of 0.7 at 734 nm. The ABTS^•+^ solution was then added to a 96-well plate. Then, 5 μL of each coffee extract was mixed with 195 μL of the working ABTS^•+^ solution and incubated for 5 min at RT. The absorbance of mixture was measured at 734 nm using a UV-Vis spectrophotometer (Thermo Scientific, Waltham, MA, USA). The results were expressed as mg of trolox equivalents per g of dry-weight coffee (mg TE/g dried sample) to calculate the IC_50_ [27]. The percentage of inhibition was calculated as follows:**% inhibition = [(A_734_ control − A_734_ test sample)/A_734_ control] × 100**

#### 2.6.2. Antibacterial Activity

Tested bacteria were prepared by transferring a single colony of *Staphylococcus aureus* ATCC2592, Methicillin-Resistant *S. aureus* (MRSA) ATCC3359, *Salmonella enteritidis* ATCC13076, *Listeria monocytogenes* NCTC7973, and *Escherichia coli* O157H7 ATCC12743 growth on Mueller–Hinton agar (MHA) into individual tubes of Mueller–Hinton broth (MHB). All strains were incubated at 37 °**C** for 24 h. Cells were harvested by centrifugation at 7000 rpm for 5 min and washed with phosphate-buffered saline (PBS). The cell suspension in PBS was adjusted to the turbidity of OD_600_ to 0.1, equal to McFarland standard number 0.5, and swabbed onto MHA [28]. Fifty microliters of coffee extracts at a concentration of 500 mg/mL were applied to each 0.8 mm filter disc and 10 μg gentamicin (Oxoid, Basingstoke, UK), and sterile distilled water was used as a positive and negative control, respectively. Inhibition zones were measured after the bacteria were incubated at 37 °C for 24 h. To determine the minimum inhibitory concentration and minimal bactericidal concentration (MIC and MBC, respectively), a microdilution method, recommended by the Clinical and Laboratory Standards Institute, was used in this study [29]. Two-fold serial dilution of the samples were carried out using MHB with a total volume of 100 μL and added to 96-well plates at a final concentration ranging from 62.5 mg/mL to 500 mg/mL. A bacterial suspension was subsequently added to obtain a final volume of 150 μL. Plates were incubated at 37 °C for 24 h, and MIC was defined as the lowest concentration of the sample at which the turbidity of sample and blank was not significantly observed at OD_600_. Suspension at the MIC value and higher concentrations were spotted on MHA using the dropped plate method and incubated at the same condition as previously mentioned. The lowest concentration at which the colony of bacteria was not detected was defined as the minimal bactericidal concentration (MBC).

### 2.7. Qualitative Examination of Fermented Coffee Properties by LC-MS and GC-MS

To preliminarily examine the roles of bacteria involved in the chemical changes in fermented coffee, LC-MS and GC-Ms were performed to detect the different compounds between fermented, control, and ordinary coffee without standard chemical compounds. The data were preliminarily analyzed by PCA plots and will be used for selection of standard chemical compounds to further analyze upscaling fermentation in the future.

#### 2.7.1. Non-Volatile Compounds

Two grams of ground coffee were placed into a 15 mL cleaned glass tube. The ground coffee was brewed with 10 mL deionized water at 90 °C for 10 min and vigorously vortexed for some time. The suspensions were centrifuged at 10,000× *g*, 4 °C for 15 min. The protein precipitation buffer (amorphous calcium phosphate; ACP) was added to supernatants to precipitate proteins before being subjected to centrifugation at the same condition. Each supernatant was filtered through the 0.2 μm membrane (Sartorius Minisart, Goettingen, Germany) and subjected to the LC-MS analysis immediately. Non-volatile compounds analysis was performed using a liquid chromatography mass spectrometry (LC-MS) instrument (Thermo Scientific, Waltham, MA, USA). Chromatographic separation was performed on a porous-shell fused-core Ascentis Express C18 (Merck, Darmstadt, Germany) analytical column (150 × 2.1 mm, particle size 2.7 μm) protected by an Ascents Express C18 guard column (0.5 cm × 2.1 mm, 2.7 μm particle size. LC-MS analyses were performed using mobile phases composed of water containing 0.1% formic acid (solvent A) and acetonitrile containing 0.1% formic acid (solvent B). The elution step was performed according to the following gradient of 5–100% B in 35 min; 100% B during 4 min; 100–5% B in 2 min; and then the column was re-equilibrated for 15 min using the initial solvent composition. Data analysis was performed using the MAVEN2 software [30].

#### 2.7.2. Volatile Compounds

To preliminarily verify the volatile compounds in fermented, non-fermented, and ordinary coffee, Gas Chromatography-Mass Spectrometry (GC-MS) was used to examine the profile of coffee odors. Two grams of each roasted coffee were ground and transferred to dry, clean glass vials, closed with plastic caps, and stored for 20 min at 60–70 °C to allow the volatile compounds to evaporate. The holder of SPME was inserted into the hollow tube so the volatile compound could be sucked into the GC-MS (Agilent model 5977, Santa Clara, CA, USA). All parameters for GC-MS protocols were set according to Andrade et al. [31]. The results were expressed as the relative percentage of each compound peak area to the total GC-MS peak area.

### 2.8. Toxicity Test in RAW 264.7 Cell

Cell cytotoxicity test was determined using 3-(4,5-Dimethylthiazol-2-yl-2,5-diphenyl-tetrazoliumbromide) (MTT) assay [32]. Briefly, murine macrophage cell line, RAW 264.7 cells, were plated at a density of 1 × 10^4^ cells/well in RPMI 1640 supplemented with 2% FBS in 96-well plates in a volume of 200 μL/well. Cells were grown at 37 °C in a 5% CO_2_/air incubator for 16–24 h before treatment with coffee extracts. LP, PM, and mixed culture fermented coffee were roasted at full city level (medium) and extracted with sterile water using the same brewing protocol. The contraction ranking from 125 μg/mL to 7.81 μg/mL was treated in the cell line for 48 h. The amount of MTT formazan was determined by measuring absorbance using a microplate reader at a wavelength of 450 nm (Biotek Instruments, Winooski, VT, USA). The percentage of viability cells was calculated by dividing the optical density (OD) of the treated cells with the (OD) from the control cells and then multiplied by 100 (as the following). For each treatment, 3 replicates were evaluated in at least 3 independent experiments.
$$\%\mathrm{cell}\;\mathrm{variability}=\frac{Absorbance\;of\;treated\;cells-background\;absorbance\;(b)}{Absorbance\;of\;untreated\;\left(c\right)-background\;absorbance\;(b)}\times 100$$
where c refers to OD value of control or non-treated cells, and b refers to blank.

### 2.9. Statistical Analysis

All measurement experiments were carried out with 3 independent repeats. The data were analyzed by using MS Excel 2021 for expression of mean ± SD of three replicates of data. One-way analysis of variance (ANOVA) and Duncan’s test were performed by using SPSS v.17 software to determine significant differences, and values were considered statistically significant if *p*  <  0.05.

## 3. Results

### 3.1. Isolation and Screening of Bacteria Based on Enzyme Activity

Two hundred and twenty-three bacteria were isolated from civet feces and screened on solid agar containing pectin, CMC, starch soluble, skim milk, and tween 80 for the capability to produce pectinase, cellulase, amylase, protease, and lipase, respectively. The primary screening results showed that 42, 62, 45, 36, and 38 isolates were detected for pectinase, cellulase, amylase, protease, and lipase production, respectively, by observing the expression of clear-zone and halos-zone, as shown in Figure 1. The colonies that expressed enzyme activities with a clear zone or halo zone greater than 2 cm were selected from both M9 and MRS agar containing each substrate for detection of total enzyme activity in M9 or MRS broth supplemented with 1% substrates, as mentioned previously (Figure 2 and Figure 3). For non-lactic acid bacteria, most of the isolates were able to produce protease and lipase, while some of those enzyme producers were able to slightly produce pectinase, cellulase and amylase (Figure 2). In contrast to lactic acid bacteria, some of the isolates were able to produce five target enzymes but with lower activity (Figure 3).

In conclusion, for the top 10 isolates obtained from preliminary screening tests based on the total enzyme activity, JT-PN06, JT-PN12, JT-PN15, JT-PN16, JT-PN20, JT-PN24, JT-PN25, JT-PN30, JT-PN37, and JT-PN39 were ranked as excellent pectinase producers, while JT-C14, JT-C22, JT-C23, JT-C25, JT-C33, JT-C37, JT-C38, JT-C46, JT-C56, and JT-C57 were able to produce cellulase. In addition, the isolate JT-A07, JT-A18, JT-A21, JT-A22, JT-A29, JT-A33, JT-A37, JT-A39, JT-A42, and JT-A43 were ranked as the best amylase-producing bacteria, whereas the isolate JT-P04, JT-P11, JT-P12, JT-P15, JT-P18, JT-P26, JT-P27, JT-P29, JT-P31, and JT-P32 were listed as protease-producing. The remaining isolates JT-L02, JT-L09, JT-L11, JT-L18, JT-L19, JT-L21, JT-L24, JT-L32, JT-L35, and JT-L38 were ranked as excellent producers of lipase. Moreover, eleven isolates, including JT-A29, JT-C14, JT-C25, JT-L19, JT-P15, JT-PN16, JT-PN06, JT-P29, JT-C38, JT-PN39, and JT-P32, exhibited several different enzyme activities (pectinase, cellulase, amylase, protease, or lipase) tested on the solid plates (Table 1) and, thus, were selected for further examination of parchment coffee digestion and 16S phylogeny identification.

### 3.2. Identification of Bacteria Producing Enzymes

For the selection of a starter culture for further coffee fermentation, non-pathogenic strains were identified and deemed as the most important criteria. Bacterial identification of selected isolates was carried out based on 16S rDNA sequence and phylogenetic analysis. The isolates obtained from M9 minimal medium, including JT-PN06, JT-PN16, and JT-C14, were grouped in the same clade with *Ochrobactrum intermedium* CCUG24694 (accession no. AM114411.1), whereas JT-A43 and JT-L19 were identified as *Acetobacter orientalis* 21F-2 (accession no. AB052706.1) and *Pseudomonas hibiscicola* ATCC19867 (accession no. AB021405.1), respectively. However, only JT-A29 and JT-P29 were clearly identified as non-pathogenic species of *Paenibacillus motobuensis* MC10 (accession no. NR043153.1) and Bacillus licheniformis DSM 13 (accession no. NR 118996.1), respectively. In addition, isolate JT-P32 was classified as *Lysinibacillus fusiformis* NBRC15717 (accession no. NR 112569.1), as shown in Figure 4.

Interestingly, the isolates of lactic acid bacteria from MRS media were discriminated into only two species of *Lactiplantibacillus plantarum* and *Enterococcus faecalis* (Figure 5). In addition, the isolates JT-PN39 and JT-C38 showed 100% similarity with *Lactiplantibacillus plantarum* CIP103151 (accession no.NR104573.1). The isolate JT-P15 showed 99% similarity with the pathogenic lactic acid bacteria, *Enterococcus faecalis* ATCC19433 (accession no. AB012212.1). For further examination of coffee digestion and fermentation, the non-pathogenic isolates, which were followed by Biosafety Guidelines for Modern Biotechnology by BIOTEC (Pathum Thani, Thailand) and NIAID Emerging Infectious Diseases/Pathogens by The National Institute of Allergy and Infectious Diseases (Annapolis, MD, USA), including isolates JT-A29, JT-P32, JT-P29, JT-PN39, and JT-C38, were selected.

### 3.3. Parchment Coffee Digetion by Selected Bacteria

To select the starter culture to be used in coffee fermentation, five isolates, including *P. motobuensis* JT-A29 and JP-P29 and *L. plantarum* JT-PN39 and JP-C38 and L. fusiformis JP-P32, were evaluated based on the ranking of their ability to produce cellulase, pectinase, amylase, protease, and lipase. Production of each enzyme by the selected strains was individually determined in minimal liquid medium by adding 1 g of sterile parchment coffee and collecting supernatants for the analysis of enzyme activity from day 0 to 8 (Figure 6). *P. motobuensis* JT-A29 revealed the maximum pectinase, cellulase, amylase, and lipase activity at 0.303 ± 0.003, 0.776 ± 0.002, 3.989 ± 0.004, and 8.929 ± 0.003 (U/g parchment coffee), respectively, at 37 °C at 48 h, while the maximum protease activity was observed at 11.260 ± 0.01 (U/g parchment coffee) at 96 h. In addition, *L. plantarum* JT-PN39 had a maximum cellulase and amylase activity of 0.579 ± 0.001 and 2.472 ± 0.006 (U/g parchment coffee), respectively, at 48 h, but the maximum pectinase, protease, and lipase activities were detected at 0.247 ± 0.003, 5.504 ± 0.006, and 6.429 ± 0.002 (U/g parchment coffee) at 48 h. Since the non-pathogenic isolates, including JT-A29 and JT-PN39, had significantly higher production of multiple enzymes at the early stage when compared to others (Table 1, Figure 2 and Figure 3), they were further tested for the fermentation of coffee in a 5 L scale.

### 3.4. Coffee Fermentation

#### 3.4.1. Bacterial Growth and pH

A small scale 5L coffee fermentation was carried out to preliminarily examine the characteristics of the starter culture during the process and to evaluate the coffee quality before industrial scaling up. A single strain of selected bacteria *P. motobuensis* JT-A29 (PM) and *L. plantarum* JT-PN39 (LP) was separately inoculated into the minimal medium containing sterile parchment coffee. In addition, we combined two strains of starter by fermenting the coffee with only PM for 24 h and then adding LP to the process at 24 h, stopping the addition after 72 h. Growth of LP in the single strain-fermenting condition reached the peak log phase at 12 h (Figure 7A), while the pH (Figure 7B) slightly dropped from pH 5 to pH 4.2 at the initiation of the process till 72 h. In addition, the stationary phase of LP in this condition was stable from 12 h to 72 h. This result suggests that this bacterium is able to continuously digest nutrients in the coffee bean, and the accumulation of lactic acid with the lower than pH 4 might have an effect on the taste of the roasted coffee. In contrast to the mixed starter culture condition (PM + LP), the maximum peak log phase of LP was delayed for 48 h (endpoint at 72 h) after inoculation at 24 h. This might be due to it competing with previously inoculated PM.

In the process that used the pure culture of PM and mixed starter culture, the growth of PM was similar, with the peak of the log phase at 18 h. However, in the mixed culture fermentation, the viability of PM immediately disappeared within 12 h of LP being added to the process. This might be caused by the effect of dropping pH to 4, which is a non-optimal growth condition for this bacterium (Figure 7A,B). In fact, during the fermentation process using a single strain of PM, the pH of the supernatant did not differ from that of the control (uninoculated) and was slightly dropped by the metabolism of the coffee bean itself.

#### 3.4.2. Enzyme Production during Coffee Fermentation

Enzyme production during coffee fermentation in 5L was studied at 0, 3, 6, 9, 12, 18, 24, 36, 48, and 72 h at 37 °C, based on the definition that one unit of enzyme activity refers to the amount of pectinase, cellulase, amylase, protease, and lipase, that is released by 1 µmol of galacturonic acid, glucose, maltose, tyrosine, and 4-nitrophenol, respectively, from 1 g of coffee bean/min at 37 °C. The maximum value of each enzyme activity among the different fermentation conditions varied, as shown in Figure 8. When using a pure or mixed starter culture, the production of all enzymes (except protease) increased along with the growth of bacteria (in Figure 7A) from 0 to 72 h. Protease activity was detected after 24 h of fermentation in all fermentation conditions. This may be because the coffee bean, as the desired enzyme substrate, contains more carbohydrates than proteins. In fact, the protease that was produced by these two strains might be synthesized at the stationary phase during coffee fermentation. Harwood and Kikuchi [33] concluded in their study that the genus *Bacillus* and related genus encode several types of proteases, and they are located within the cell, such as in the cytoplasm, cell membrane, cell wall, and external milieu. These proteases are not essential for growth, but they are useful for survival in a competitive environment. Moreover, protease helps control specific aspects of metabolism and cellular behavior.

### 3.5. Sensory Evaluation

Six different sensory analyses were carried out to describe the flavor characteristics of 5 L roasted, fermented coffee in this study. The cupping test score (Table 2) was used to characterize some attributes of each coffee, including ordinary coffee (OC), control or non-fermented (C; uninoculated), *P. motobuensis* JT-A29 (PM), *L. plantarum* JT-PN39 (LP), and mixed starter cultures (PM + LP) fermented coffee. Civet coffee (CC), the most expensive coffee, was used as the animal fermented coffee representation. LP fermented coffee expressed the highest score for all attributes, while non-fermented coffee or control showed the lowest score. These findings imply that LP might help improve the quality of the coffee fermentation process and the flavor quality to generate specialty coffee. However, due to the acid production during fermentation, it could reduce the body attribute quality. The utilization of PM strain alone caused the reduction in coffee quality, but using mixed starter culture increased the quality of flavor when compared to the control and ordinary coffee. The flavors and tastes of the fermented coffee were characterized by SCAA-certified Q-graders, as shown in Table 3.

### 3.6. Biological Functions of Roasted Fermented Coffee

#### 3.6.1. Total Phenolic Content and Antioxidants

The total polyphenol content (TPC) was measured based on a redox reaction, as presented in Table 4. Roasted fermented coffee using PM process showed the highest amount of polyphenol content at 84.42 ± 0.51 mg GAE/g coffee, while LP fermentation had 83.95 ± 0.47 mg GAE/g coffee. In addition, green bean fermented coffee had a lower amount of TPC, and LP fermentation had the greatest amount among green bean coffee. The ABTS radical scavenging activity of different concentrations of each coffee was measured, and the IC_50_ was determined (Table 4). The IC_50_ can be defined as the concentration of the coffee extracts required to inhibit 50% of radicals, and lower IC_50_ values have better antioxidant activity. Similar to the TPC value, the antioxidant activity of roasted, fermented coffee and green bean was revealed to be the highest in PM and LP fermentation, respectively. Overall, fermented coffee showed better health benefits and antioxidant properties than ordinary coffee, non-fermented, civet coffee, and commercial coffee.

#### 3.6.2. Antipathogenic Bacterial Activity

The anti-pathogenic foodborne bacteria activity of the five species mentioned previously was determined using the disc diffusion method. LP fermentation process had, among the fermented coffee and non-fermented, the highest significant inhibitory effect on Gram-positive pathogenic bacteria, including *S. aureus*, *S. aureus* (MRSA), and *L. monocytogenes*. Moreover, fermented coffee using a single species of LP or mixed culture was able to inhibit both Gram-negative and Gram-positive pathogenic bacteria with a better value than that of PM. All fermented types of coffee also revealed better bactericidal activity than non-fermented and commercial coffee (Table 5). The results of minimum inhibitory concentration (MIC) and minimal bactericidal concentration (MBC) are shown in Table 6.

### 3.7. Non-Volatile and Volatile Compound Qualification in Fermented Coffee

The LC-MS chromatographic profiling of coffee in 5 L fermentation was observed qualitatively without comparison with standard compounds in this study. Thirty-five compounds were detected (Appendix A), and LP fermentation process expressed significantly higher chlorogenic acid, trigonelline, ferulic acid, lactic acid, succinic acid, catechins, and shikimic acid content, while caffeine was lower. A principal component analysis (PCA) was carried out to compare the differentiation of non-volatile profiles among the non-fermented and fermented coffee. The PCA analysis with semi-quantitative data of each compound with a maximum peak area larger than 10^5^ was used for calculation. Figure 9 clearly shows the use of each starter culture involved in metabolisms inside coffee beans during the fermentation. Interestingly, ordinary coffee and civet fermented coffee did not exhibit any difference in coffee metabolism. However, the results of volatile compounds were found to be different; the civet fermented coffee seemed to generate similar compounds to those of the control fermented coffee. These findings suggest that the metabolism inside a coffee bean in a wet condition (i.e., intestine and liquid M9 medium) might affect the formation of some aromas, such as Pyrazine, 2-ethyl-6-methyl, 3,5-Dimethylcyclohex-1-ene-4-carboxaldehyde, and Pyrazine, 2-ethyl-3,5-dimethyl (Figure 10). The list of volatile compounds detected in all types of coffee in this study is presented in Appendix A.

### 3.8. Toxicity of Fermented Coffee

To evaluate whether the fermented process results in the formation of toxic residues, an MTT assay was conducted to measure the cell viability of RAW 264.7 cells. Non-fermented, ordinary coffee and fermented coffee were extracted with sterile milliQ water at concentrations ranging from 500 to 31.25 μg/mL, and quercetin, a rich natural compound in coffee beans, was used as the positive control. Every concentration in all coffee extracts had a viability percentage above 80%, which indicated it was non-toxic (Figure 11). However, the results observed in this study were only the preliminary test, and an in vivo examination may be necessary to discriminate acute and chronic toxicity.

## 4. Discussion

Fermentation is a process during which macromolecules are broken down into small molecules, other products, and gases. It has been introduced to improve flavors and aromas. Mostly, fermentation is carried out using ripened coffee fruits, or coffee cherries, which are immediately fermented after harvesting [34]. The coffee industry employs three different methods, including dry, wet, and semidry processes, to generate novel flavors. These processes are considered to be time- and cost-consuming and are unreliable in terms of consistency of flavor in each batch processed. In recent years, the selection of starter cultures of bacteria and yeasts have been combined with normal flora fermentation in coffee manufacturing [35,36]. The purpose of using the starter culture is mainly to remove the coffee mucilage layer, such as pectin and other polysaccharides, from the parchment coffee [36,37]. The fermentation process is facilitated by enzymes from microorganisms (yeast, bacteria, and fungi) that naturally reside on coffee fruits and in the environment to degrade carbohydrate composition [38]. Although there are several commercially available enzymes for coffee fermentation, microorganisms play a major role in generating unique metabolites that help establish the development of aroma and flavors. For these reasons, this study aimed to isolate the civet’s gut bacteria for utilization in coffee fermentation. From unpublished data on civet’s gut microbiome, lactic acid bacteria were found to be the minority population, but most of the bacteria were found to belong to the pathogenic genera. However, several strains of *Lactiplantibacillus plantarum* were isolated with a varity of pectinase, cellulase, amylase, lipase, and protease production. Another non-pathogenic species of *Paenibacillus motobuensis* was also selected due to its ability to produce enzymes. Peñuela et al. [39] reported the use of several species of lactic acid bacteria, including *Lactiplantibacillus* spp., *Lactococcus* spp., and *Leuconostoc* spp. in the process of coffee fermentation; however, there has been no report on the utilization of *P. motobuensis.* Moreover, only a few studies used parchment coffee in fermentation processes. Many studies used *Lactiplantibacillus* and *Lactococcus* for green coffee bean fermentation [40]: however, there is no report about their enzyme production that may be involved in coffee bean digestion. Thus, our study was the first report to demonstrate that the selected starter culture with metabolisms changes could provide outcomes similar to those of using synthetic enzymes in coffee fermentation.

Sensory analysis of coffee beverages was determined by five certified experts or Q-Grader using the cupping test method, following the Specialty Coffee Association of America Cupping Protocols [22]. The evaluated scores of overall fermented coffee and aroma were more acceptable. The fermented coffee with LP expressed a very good smell, such as caramel, herb, fresh leaf, and floral. The characteristic aroma of the LP fermentation correlated well with previous studies. According to Pereira et al. [41], several lactic acid bacteria, including LP, produce the flavor of active ester compounds and lactic acid. Nevertheless, modifications of coffee flavor constituents during parchment coffee bean fermentations by LP showed that inoculation influenced coffee quality since its non-volatile and volatile compounds were significantly different from those of others. Moreover, the compounds found in LP fermentation were significantly different in the amount of chlorogenic acid, trigonelline, ferulic acid, lactic acid, succinic acid, catechins, and shikimic acid. And this finding was in agreement with a previous study reporting that higher levels of malic acid and succinic acids gave rise to sweet and acidic flavors [1,6]. However, caffeic acid concentrations in coffee fermented with LP were reduced. Lactic acid accumulation at a higher level could be associated with several pathways, such as pyruvate via glycolysis, which converts glucose and fructose mostly to lactic acid [42]. In correspondence with a previous study, supplemented glucose in the coffee fermentation resulted in significantly enhanced microbial growth and acid production during LP coffee bean fermentation [1,41].

The volatile compounds of coffee beans develop primarily in the form of alcohol, acids, ester, and aldehydes [43]. However, some of these flavor-active compounds originate from the bean itself. On the other hand, several compounds were formed by microbial-derived metabolites from starter cultures that can also diffuse into the coffee during the fermentation process. This research showed an increase in the volatile profile and sensory attributes of the fermented coffee using LP, such as acetic acid, furfural, 5-methyl furfural, 3,5-dimethylcyclohex-1-ene-4-carboxaldehyde, maltol, 10-hydroxydecanoic acid, limonen-6-ol, pivalate, tetradecane, 2,6,10-trimethyl-, α-furfuryliden-α-furylmethylamine, 4-vinylguaiacol, 5-methyl-2-phenyl-2-hexenal, and megastigmatrienone. The higher contents of furfurals, furans, and maltol and lower levels of substituted pyrazines in LP fermentation could possibly be a source of its stronger caramel, almond, nutty, and sweet flavors.

The introduction of LP to coffee fermentation resulted in an increase in the TPC and antioxidant activity [44]. This finding was similar to our result presented in Table 4. In addition, phenolic compounds, which are responsible for the astringency, flavor, and antioxidant properties of coffee [45], were found to be chlorogenic acid, gallic acid, and caffeic acid [46,47]. The antioxidant potency of these polyphenol compounds in coffee revealed that fermented green coffee had higher activity than that of roasted ones. According to ABTS assays, the roasted, fermented coffee with LP exhibited increased antioxidant activity, and this result was consistent with the previous studies by Haile and Kang [48] and Kwak et al. [35]. During fermentation, proteolytic enzymes from the starter culture hydrolyze the complexes of phenolics into soluble-free phenols and more biologically active functions [49,50]. Moreover, antioxidant activities in light, medium, and dark roasted coffee showed a decrease in antioxidant activities. This could be caused by the degradation of chlorogenic acids, along with other non-volatile phenolic derivatives, which are significant contributors to the antioxidant activities of green coffees during roasting [51]. It is well known that roasting greatly affects the chemical composition of coffee beans due to high temperatures [52,53,54]. For example, chlorogenic acids are broken down, and other metabolites are transformed into non-antioxidant compounds [51,55,56,57].

The coffee fermented with LP revealed antibacterial activity where Gram-positive bacteria, including *S. aureus*, *S. aureus* (MRSA), and *L. monocytogenes,* are more susceptible than Gram-negative bacteria, such as *E. coli* and *S. enteritidis*. The results were in agreement with previous data obtained using brewed coffee [58,59,60]. Moreover, dark-roasted coffee had more inhibitory activity than medium- and light-roasted levels. However, the efficiency of fermented coffee with LP against each bacterium was different because of many factors, such as the Gram-positive and Gram-negative bacteria cell structure, their specialization in the outer membrane envelope, and the active compounds of the extract. Normally, hydrophobic compounds (phenols and tannins) are difficult to uptake into the Gram-negative bacteria’s outer membrane [58].

In conclusion, the fermentation of parchment coffee with LP had a more positive impact than that of PM and mixed starter culture for several reasons, including flavors, aroma, non-toxic, and health benefits. Together with its safety approval, LP may be promising as the potential starter culture for the innovative process of coffee fermentation and may be further developed as a functional beverage.

## 5. Conclusions

This study aimed to evaluate the uses of gut bacteria from civets for utilization in coffee fermentation. The innovative fermentation process was formulated based on the enzyme production by selected bacteria in accordance with the safety for food application. Non-pathogenic isolates *Lactiplantibacillus plantarum* JT-PN39 (LP) and *Paenibacillus motobuensis* JT-A29 (PM), which were able to produce several target enzymes to digest the nutrients in coffee beans, were selected and subjected to 5L parchment coffee fermentation. Examination of the bacterial growth and enzyme production during the fermentation revealed the short period of further large-scale coffee fermentation and may help reduce the time consumption. In addition, this study aimed to improve the uncertainty of the quality of coffee produced via fermentation via the classical method of wild strains or the normal flora of bacteria and yeasts on coffee cherries. Moreover, this method was designed to be able to sustain year-round coffee fermentation and to replace the classical method, where fermentation is available only during the coffee cherry harvesting season.

The benefits of fermented coffee were also determined for anti-pathogenic bacteria and antioxidant activity, both of which were significantly higher in fermented coffee than in non-fermented coffee and commercial coffee. In fact, the qualitative chemical composition of fermented coffee tended to increase the accumulation of beneficial compounds, such as chlorogenic acid and trigonelline. These compounds are well known for their anti-inflammation and immunological stimulation. However, the quantitative determination of the non-volatile and volatile compounds must be evaluated in the future. For both quality of flavor, aroma, and the benefit value, LP fermented coffee is a promising process with a viable use in industrial fermented coffee. In addition, this species is a well-known GRAS strain.

## Figures and Tables

**Figure 1 foods-12-02894-f001:**
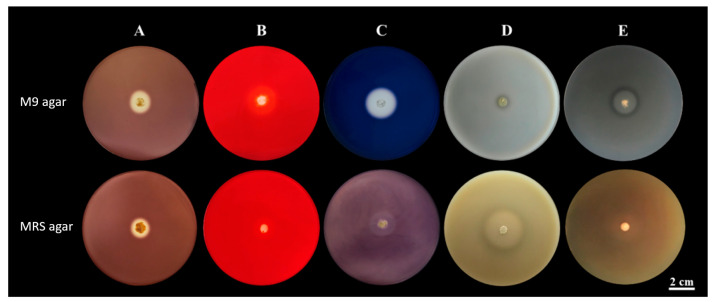
The representative of solid media for primary screening of pectinase, cellulase, amylase, protease, and lipase production using 1% of pectin (**A**), CMC (**B**), starch (**C**), skim milk (**D**), and tween 80 (**E**) as the substrate, respectively.

**Figure 2 foods-12-02894-f002:**
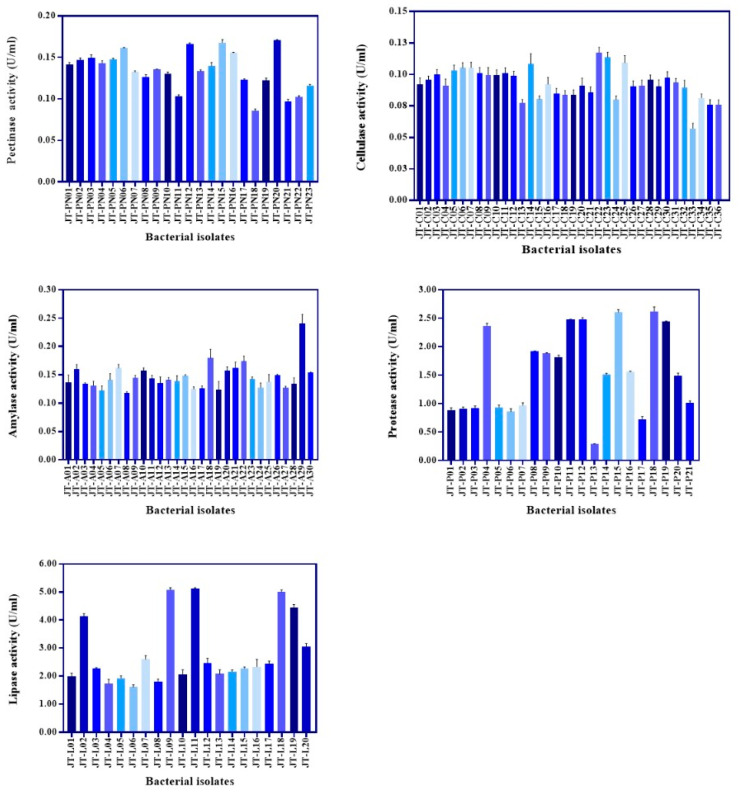
Total enzyme activity of the isolates grown in M9 broth supplemented with 1% of individual substrate for pectinase, cellulase, amylase, protease, and lipase, respectively. The calculation of each enzyme activity was referred to as unit/mL (U/mL) when the bacteria were cultivated at 37 °C for 24 h, and the supernatant was collected at the end of the cultivation time for enzyme detection.

**Figure 3 foods-12-02894-f003:**
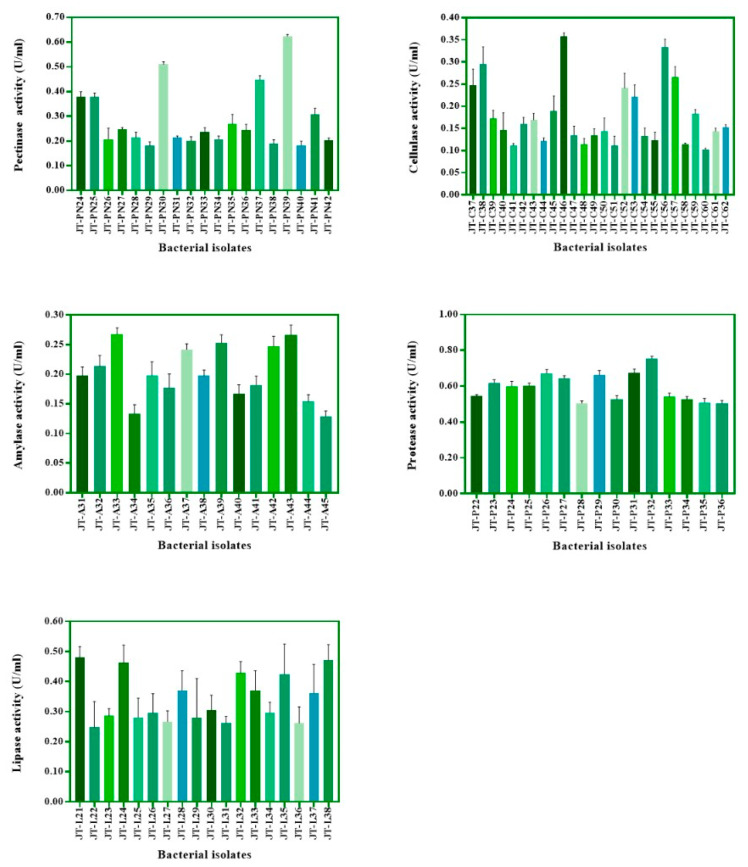
Total enzyme activity of the isolates grown in MRS broth supplemented with 1% of individual substrate for pectinase, cellulase, amylase, protease, and lipase, respectively. The calculation of each enzyme activity was referred to as unit/mL (U/mL) when the bacteria were cultivated at 37 °C for 24 h, and the supernatant was collected at the end of the cultivation time for enzyme detection.

**Figure 4 foods-12-02894-f004:**
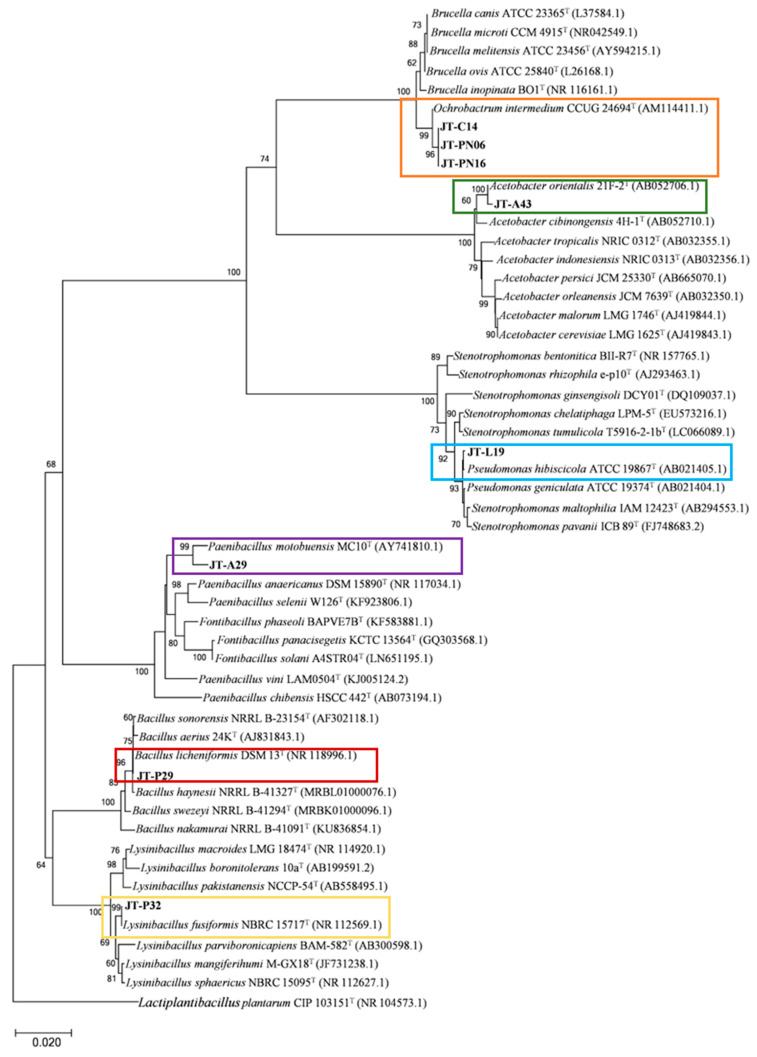
Identification of the bacterial isolates, which were able to produce several target enzymes on the M9 medium with supplemented substrates using 16S rDNA phylogeny. A phylogenetic tree was constructed using the NJ method. Numerals at the nodes indicate bootstrap values (%) derived from 1000 replications. The isolates used in this study are surrounded by color boxes.

**Figure 5 foods-12-02894-f005:**
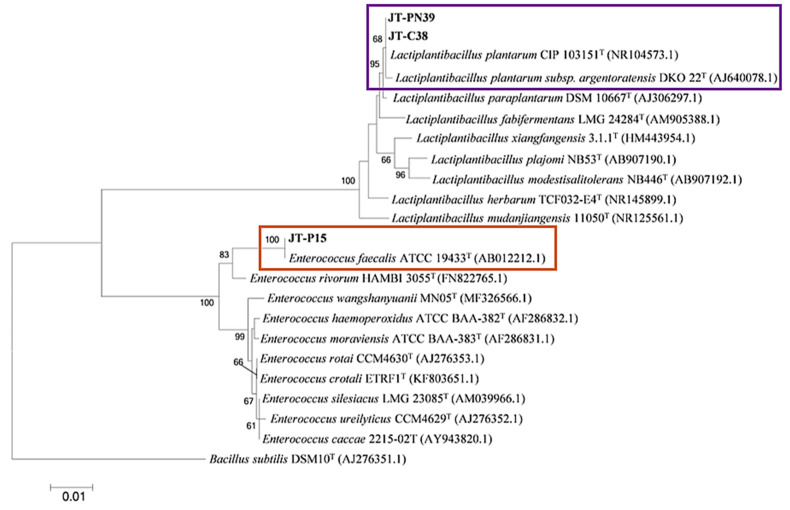
A phylogenetic tree based on 16S rRNA gene sequences of lactic acid bacteria that were able to produce several target enzymes raking at the top three. Numerals on the nodes indicate bootstrap values (%) derived from 1000 replications. The isolates used in this study are surrounded by color boxes.

**Figure 6 foods-12-02894-f006:**
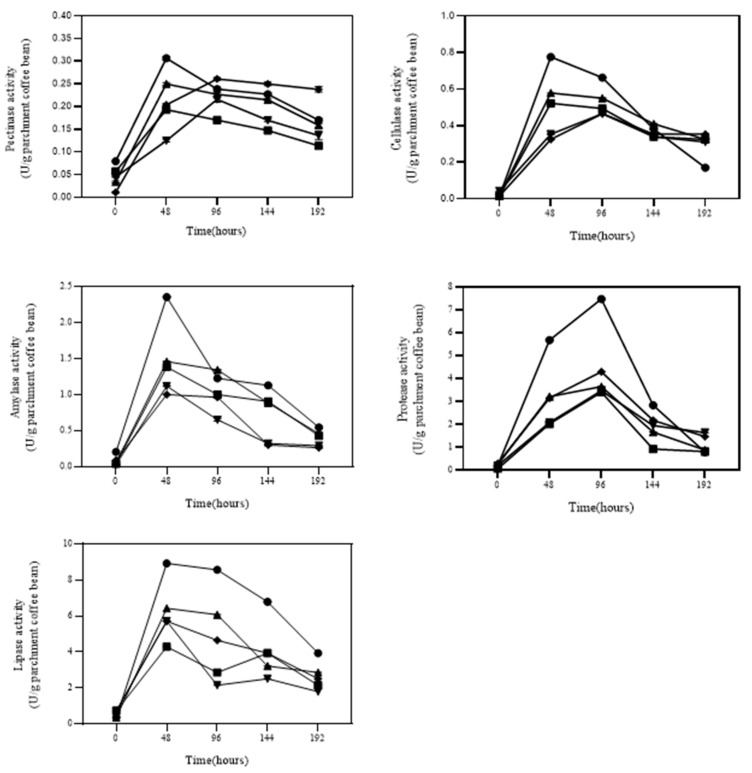
Digestion of parchment coffee in minimal medium supplemented with 1 g coffee bean and examination of the enzymes production by five selected isolates including JT-A29 (
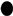
), JT-PN39 
(
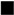
), JT-C38 (
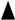
), JT-P29 (
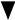
), and JT-P32 (
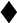
) from 0 to 192 h.

**Figure 7 foods-12-02894-f007:**
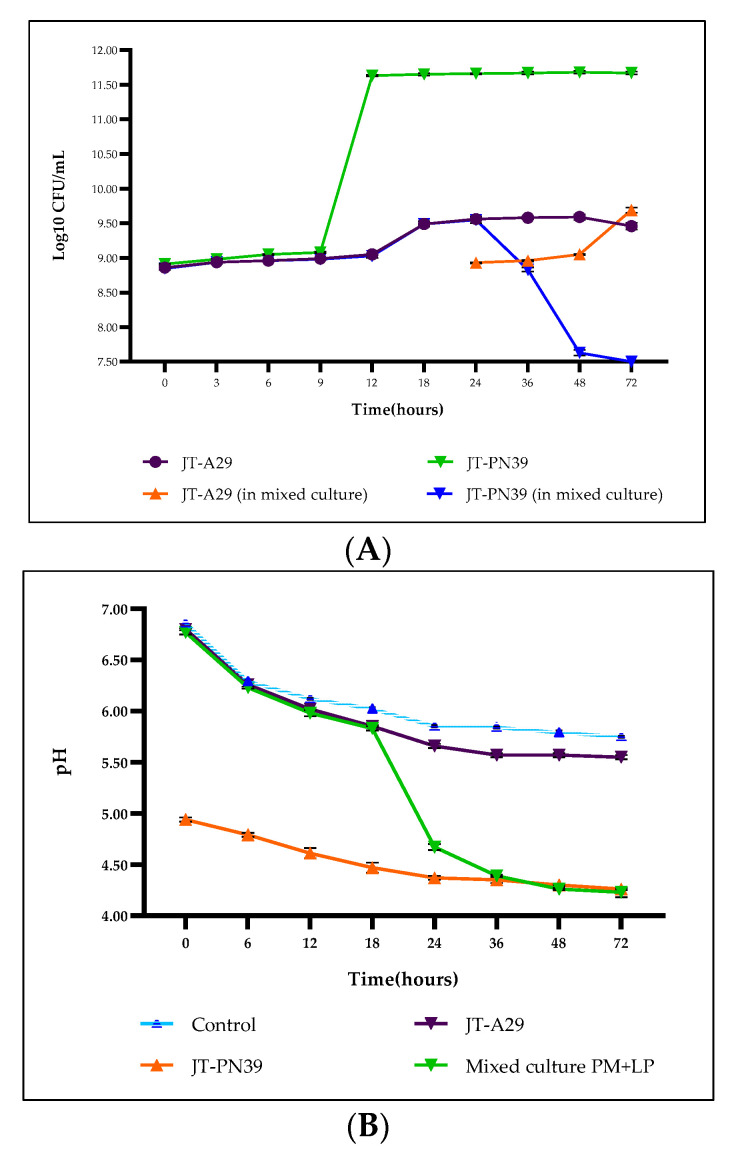
Growth of starter culture *P. motobuensis* JT-A29 (PM) and *L. plantarum* JT-PN39 (LP) during coffee fermentation in 5 L. Growth (**A**) and variation in pH (**B**), which refer to acid production by bacteria, were examined starting from the initial process till 72 h under the uses of pure starter and mixed starter culture fermentation.

**Figure 8 foods-12-02894-f008:**
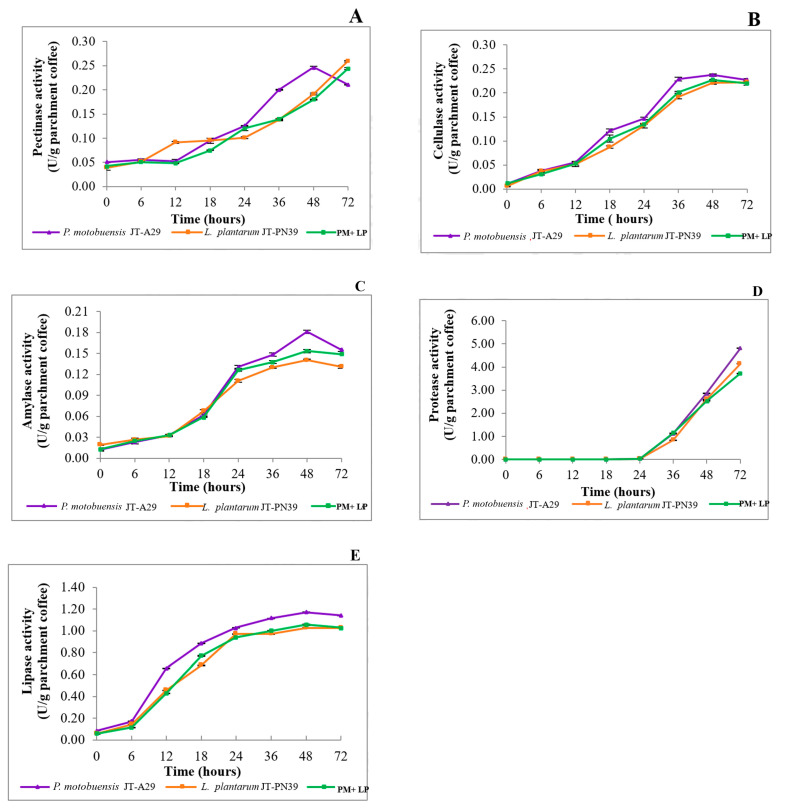
Production of five target enzymes (**A**–**E**) by starter culture *P. motobuensis* JT-A29 (PM), *L. plantarum* JT-PN39 (LP), and mixed culture PM + LP during 5L coffee fermentation from 0 to 72 h.

**Figure 9 foods-12-02894-f009:**
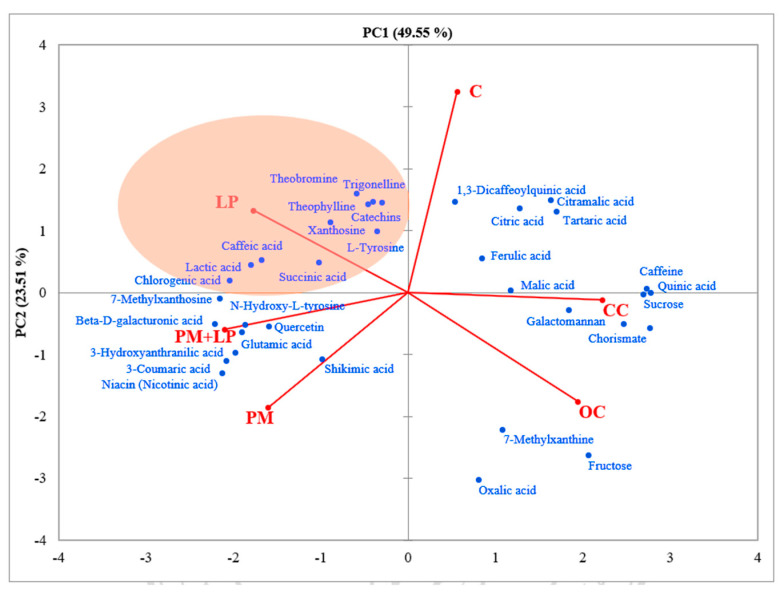
Semiquantitative component analysis for non-volatile compounds in roasted coffees. The principal component analysis (PCA) was performed with XLSTAT 19.02 (Addinsoft, New York, NY, USA) using the peak area of LC-MS detection.

**Figure 10 foods-12-02894-f010:**
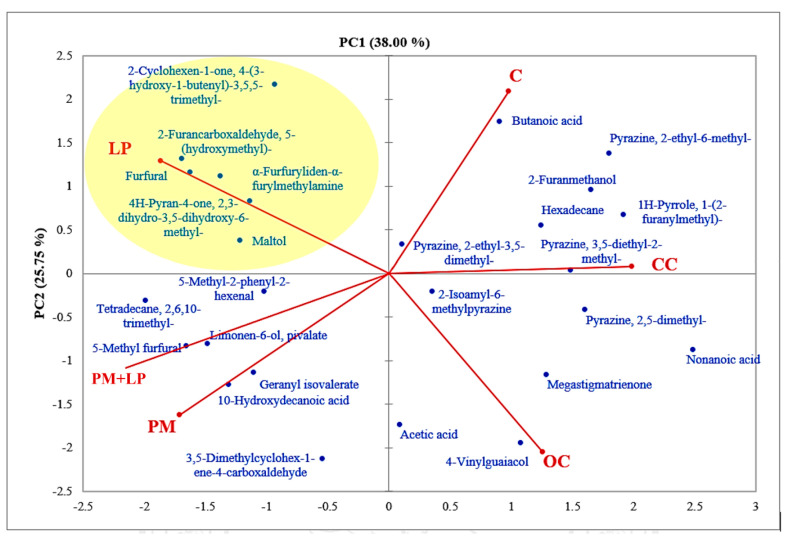
Semiquantitative component analysis for volatile compounds in roasted coffees. The principal component analysis (PCA) was performed with XLSTAT 19.02 (Addinsoft, New York, NY, USA) using the peak area of GC-MS detection.

**Figure 11 foods-12-02894-f011:**
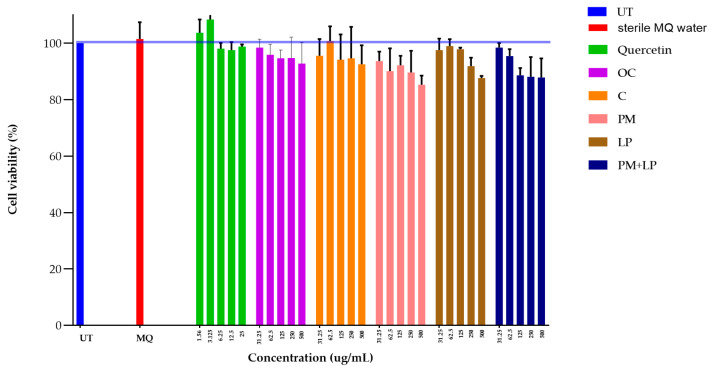
Cell viability of RAW 264.7 cells after 48 h of treatment with coffee extracts at various concentrations (500 μg/mL to 31.5 μg/mL). The viability of the untreated (UT) cells was set as 100% viability, while milliQ water and quercetin were set as the negative and positive control, respectively.

**Table 1 foods-12-02894-t001:** Selected isolates capable of producing several enzymes.

Isolates	Cellulase	Pectinase	Amylase	Protease	Lipase
**JT-A29**	**+**	**+**	**+**	**+**	**+**
JT-C38	+	+	−	−	+
**JT-PN39**	**+**	**+**	**+**	**+**	**+**
JT-A43	+	+	+	−	−
JT-C14	+	+	−	−	−
JT-L19	−	−	+	+	+
JT-P15	−	−	+	+	−
JT-PN06	−	+	+	+	+
JT-PN16	−	+	−	−	+
**JT-P29**	**+**	**+**	**+**	**+**	**+**
**JT-P32**	**+**	**+**	**+**	**+**	**+**

(+) Positive or present and (−) Negative or absent. Selected strains for further experiments are shown in bold letters in the table.

**Table 2 foods-12-02894-t002:** Coffee sensory analysis based on the SCAA cupping test.

Attributes and Varieties	Coffee					
	OC	C	PM	LP	PM + LP	CC
Fragrance/Aroma	7.00	7.00	7.00	7.25	7.00	7.00
Acidity	7.00	7.00	7.00	7.25	7.25	7.00
Flavor	7.00	7.00	7.00	7.25	7.00	7.00
Body	7.00	7.00	6.75	7.25	6.75	7.00
Aftertaste	7.00	7.00	7.00	7.00	7.00	7.00
Uniformity	10.00	10.00	10.00	10.00	10.00	10.00
Balance	7.00	6.25	7.00	7.25	7.00	7.00
Clean cup	10.00	10.00	10.00	10.00	10.00	10.00
Sweetness	10.00	10.00	10.00	10.00	10.00	10.00
Overall	7.00	7.00	6.75	7.00	7.25	7.00
**Total score**	79.00	78.25	78.50	80.25	79.25	79.00

**Table 3 foods-12-02894-t003:** Aroma of fermented coffee evaluation by SCAA-certified Q-graders.

Coffee	Aroma	
	Dry Aroma	Wet Aroma
**OC**	roasted, dark chocolate, cocoa nibs roasted	meat, caramel, brown sugar, smoke, hint of jackfruit
**C**	mozzarella cheese, spicy	cheese, cherry, mint, hint of lemon
**PM**	musty, nutty, dry wood, hint cheese	Musty, ferment, peach, hint of floral, hint cheese
**LP**	tobacco, cigarette, dry wood	caramel, herb, fresh leaf, flower, citrus, yoghurt
**PM + LP**	tobacco, dry wood, spicy	spicy, tea, chocolate, butter, seaweed
**CC**	rose, pinecone seed, dry wood	brown sugar, caramel, spicy, dry wood

**Table 4 foods-12-02894-t004:** Total polyphenol contents and antioxidant activity of coffee extracts.

Coffee Extracts	Antioxidant Activity	
	ABTS Assay IC_50_ (mg/mL)	TPC (mg GAE/g Coffee)
** *Roasted Coffee Beans* **		
Ordinary coffee (OC)	1.046 ± 0.006 ^de^	79.72 ± 0.54 ^de^
Non-fermented (C)	1.068 ± 0.008 ^f^	78.50 ± 0.72 ^f^
**PM**	**0.999 ± 0.006 ^a^**	**84.42 ± 0.51 ^a^**
**LP**	**1.000 ± 0.002 ^a^**	**83.95 ± 0.47 ^a^**
PM + LP	1.019 ± 0.017 ^bc^	82.70 ± 0.78 ^b^
Civet coffee (CC)	1.072 ± 0.006 ^fg^	78.38 ± 0.71 ^f^
Civet coffee commercial	1.104 ± 0.018 ^ij^	76.19 ± 0.31 ^h^
Commercial Coffee 1	1.117 ± 0.010^j^	68.50 ± 0.67 ^i^
Commercial Coffee 2	1.005 ± 0.006 ^ab^	80.07 ± 0.73 ^de^
Commercial Coffee 3	1.024 ± 0.009 ^c^	79.95 ± 0.35 ^de^
** *Green coffee bean* **		
Ordinary coffee (OC)	1.060 ± 0.010 ^ef^	79.33 ± 0.71 ^def^
Non-fermented (C)	1.087 ± 0.002 ^gh^	79.13 ± 0.42 ^ef^
PM	1.058 ± 0.008 ^ef^	80.23 ± 0.30 ^d^
**LP**	**1.041 ± 0.004 ^d^**	**81.87 ± 0.36 ^bc^**
PM + LP	1.059 ± 0.009 ^ef^	81.64 ± 0.18 ^c^
Civet coffee (CC)	1.097 ± 0.007 ^hi^	77.33 ± 0.76 ^g^

Mean of triplicate in each column bearing the same letters is not significantly different (*p* > 0.05) from one another using Duncan’s Test (mean ± standard variation). Fermented coffee with the highest antioxidant activity and total phenolic content are shown in bold letters in the table.

**Table 5 foods-12-02894-t005:** Inhibition zone of coffee extracts to the representative of pathogenic bacteria. Chloramphenicol 10 μg was used as the positive control.

Coffee Extract	Zone of Inhibition (mm)				
	*S. aureus*	*S. aureus* (MRSA)	*L. monocytogenes*	*E. coli*	*S. enteritidis*
Chloramphenicol	19.50 ± 0.50 ^a^	7.00 ± 0.00 ^h^	24.44 ± 0.51 ^a^	15.89 ± 0.84 ^a^	20.56 ± 0.51 ^a^
** *Roasted coffee beans* **					
Ordinary coffee (OC)	11.89 ± 0.10 ^f^	10.11 ± 0.10 ^g^	13.00 ± 0.00 ^h^	0.00 ± 0.00	0.00 ± 0.00
Non-fermented (C)	14.17 ± 0.17 ^d^	24.83 ± 0.44 ^bc^	12.78 ± 0.38 ^h^	0.00 ± 0.00	0.00 ± 0.00
PM	12.11 ± 0.10 ^f^	24.39 ± 0.25 ^c^	15.22 ± 0.19 ^d^	0.00 ± 0.00	0.00 ± 0.00
**LP**	19.00 ± 0.17 ^b^	29.72 ± 0.25 ^a^	16.50 ± 0.44 ^b^	10.78 ± 0.38 ^b^	10.89 ± 0.10 ^b^
PM + LP	18.06± 0.25 ^c^	25.56 ± 0.96 ^b^	15.89 ± 0.25 ^c^	10.50 ± 0.50 ^b^	10.78 ± 0.25 ^b^
Civet coffee (CC)	11.89 ± 0.35 ^f^	18.44 ± 0.38 ^f^	14.33 ± 0.33 ^ef^	10.22 ± 0.38 ^b^	10.00 ± 0.00 ^c^
Commercial civet coffee	13.28± 0.10 ^e^	22.83 ± 0.50 ^d^	14.56 ± 0.51 ^e^	9.17 ± 0.17 ^c^	9.33 ± 0.00 ^d^
Commercial Coffee 1	11.33 ± 0.29 ^g^	21.72 ± 0.48 ^e^	12.78 ± 0.25 ^h^	0.00 ± 0.00	0.00 ± 0.00
Commercial Coffee 2	9.44 ± 0.51 ^h^	24.56 ± 0.38 ^c^	13.78 ± 0.25 ^fg^	0.00 ± 0.00	0.00 ± 0.00
Commercial Coffee 3	11.86 ± 0.13 ^f^	22.56 ± 0.38 ^d^	13.22 ± 0.19 ^gh^	0.00 ± 0.00	0.00 ± 0.00

Mean of triplicate in each column bearing the same letters is not significantly different (*p* > 0.05) from one another when using Duncan’s Test (mean ± standard variation).

**Table 6 foods-12-02894-t006:** MIC and MBC value of coffee extracts with pathogenic bacteria.

Coffee Extracts	Bacteria									
	*S. aureus*		*S. aureus* (MRSA)		*L. monocytogenes*		*E. coli*		*S. enteritidis*	
	MIC (mg/mL)	MBC (mg/mL)	MIC (mg/mL)	MBC (mg/mL)	MIC (mg/mL)	MBC (mg/mL)	MIC (mg/mL)	MBC (mg/mL)	MIC (mg/mL)	MBC (mg/mL)
Chloramphenicol	0.004	0.007	0.004	0.007	0.004	0.007	0.004	0.007	0.004	0.007
** *Roasted coffee beans* **										
Ordinary coffee (OC)	41.67	166.67	41.67	83.33	41.67	166.67	ND	ND	ND	ND
Non-fermented (C)	41.67	166.67	41.67	83.33	83.33	>166.7	ND	ND	ND	ND
PM	41.67	166.67	41.67	83.33	41.67	166.67	ND	ND	ND	ND
**LP**	41.67	83.33	41.67	83.33	41.67	83.33	41.67	83.33	41.67	83.33
PM + LP	41.67	83.33	41.67	83.33	41.67	166.67	41.67	166.67	41.67	166.67
Civet coffee (CC)	83.33	>166.7	41.67	83.33	41.67	166.67	166.67	166.67	166.67	166.67
Commercial civet coffee	41.67	166.67	41.67	166.67	41.67	166.67	166.67	166.67	166.67	166.67
Commercial Coffee 1	41.67	166.67	41.67	166.67	41.67	166.67	ND	ND	ND	ND
Commercial Coffee 2	83.33	>166.7	41.67	166.67	41.67	166.67	ND	ND	ND	ND
Commercial Coffee 3	83.33	>166.7	41.67	83.33	83.33	>166.7	ND	ND	ND	ND

ND = not detected.

## Data Availability

The data presented in this study are available on request from the corresponding author or by downloading the Appendix A Ngamnok T. Fermented coffee. (https://1drv.ms/f/s!Ah6GbIYZhQuQgYtN2QAafQ1D0569FA?e=bcQoVj, accessed on 6 June 2023).

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
