# Peer review of "Efficiency of Lactiplantibacillus plantarum JT-PN39 and Paenibacillus motobuensis JT-A29 for Fermented Coffee Applications and Fermented Coffee Characteristics"

_foods, 2023, doi:10.3390/foods12152894_

Round 1
Reviewer 1 Report
Dear Editors and authors,
1-Abstract of the manuscript needs to be rewritten and needs to add some important results in the study.
2-The introduction of the manuscript should contain a chapter on the addition of probiotic bacteria in plant foods and their effects. I suggest to add
Al-Sahlany, S. T. G., Khassaf, W. H., Niamah, A. K., & Abd Al-Manhel, A. J. (2023). Date juice addition to bio-yogurt: The effects on physicochemical and microbiological properties during storage, as well as blood parameters in vivo. Journal of the Saudi Society of Agricultural Sciences, 22(2), 71-77.
Küçükgöz, K., & TrzÄ…skowska, M. (2022). Nondairy probiotic products: Functional foods that require more attention. Nutrients, 14(4), 753.
3-Many of the work methods do not contain scientific references, so it is difficult for the reader to return to the original work method. See line 93-108, 127-143, 158-168, and ..................etc.
4-The authors did not mention the reason for selecting the two bacterial isolates among the other isolates, what is the scientific basis for selecting the bacterial isolates. (P. motobuensis JT-A29 (PM) and L. plantarum JT-PN39).
5-Table 1 The word cellulase enzyme is wrong.
6-Figure 7a is not clear, did the bacterial isolate (JT-A29) grow after 24 hours, while it was noted that it decreased the pH from zero time (Figure 7 b).
7-Figures 8 and 9 for the results of the flavor compounds are not clear to the reader. It is preferable that these figures be in tables form, which makes it easier to read.
The language of the manuscript is good and easy for the reader.
Sridharan, S., & Das, K. M. S. (2019). A study on suitable non dairy food matrix for probiotic bacteria–a systematic review. Current Research in Nutrition and Food Science Journal, 7(1), 05-16.
Author Response
Dear Reviewer,
We would like to thank you for your valuable comments and suggestions. We provide more information or answer the comments in this report as below. For edited manuscript upon your suggestions, the manuscript was adding more detail and references and revised at the position which was highlighted with the yellow color. However, the line number may be changed from the original comments so please find at the highlight sentences.
Sincerely yours,
Jomkhwan

Reviewer 2 Report
This Manuscript deals with an interesting scientific issue, the planning of the research, conducting the experiments, as well as the obtained results do not raise any objections, nevertheless, I have a few suggestions for the Authors as follows:
Introduction: lines 41-50; 56-65 - please support these claims with relevant literature items.
In section: Statistical analysis you wrote that Tukey tests were performed to determine significant differences, and in the legend to tables 4 and 5 you refer to the Duncan's test - please decide which of the post-hoc tests was performed.
Author Response

(The authors gave the same response as above.)

Reviewer 3 Report
The manuscript reported a well-designed and thoroughly executed study. The authors have done an excellent job outlining their methods and providing detailed explanations of the results. I was particularly impressed with the clarity and thoroughness of the data presented. Overall, this paper is a valuable contribution to the field, and I look forward to seeing further research from these authors. Some itemized comments are listed below:
L76-79: Please provide more details or examples of diseases that can be prevented. Also, please provide references.
L325: Add a comma between 62 and 45.
Most graphs do not have error bars!
L529-531: Please provide a reference for this sentence. Protease might be a secondary metabolite of the strains, which was synthesized in the stationary phase.
L547: mixed
Author Response

(The authors gave the same response as above.)

Round 2
Reviewer 1 Report
Dear Editors,
The authors have made modifications and corrections to the manuscript. I think the manuscript is ready for publication.
The language of the manuscript is good and simplified for the reader.